# A Framework for Developing Awareness Interventions: A Case of Mobile Bullying

**DOI:** 10.3390/ijerph22050774

**Published:** 2025-05-14

**Authors:** Portia Mathimbi

**Affiliations:** Department of Information Systems, University of Cape Town, Rondebosch, Cape Town 7700, South Africa; bthpor003@myuct.ac.za

**Keywords:** cyberbullying, mobile bullying, raising awareness, awareness intervention, prevention programs

## Abstract

Cyberbullying is a pervasive social issue that has gained increasing attention due to its serious consequences, such as cyberbullicide, which is suicide resulting from cyberbullying. Researchers have called for various interventions and the involvement of multiple stakeholders, including psychologists and law enforcement agencies, to address the problem. Recent studies have shown that the majority of cyberbullying incidents occur on mobile devices, referred to as mobile bullying. Therefore, there is a need to focus intervention efforts on mobile bullying, which is a subset of cyberbullying. The objective of this study was to develop a framework that could guide the creation of an awareness intervention aimed at raising awareness of mobile bullying. Although several meta-analyses have been conducted over the years on intervention effectiveness. There is a gap from the lack of frameworks and requirements to guide the development of awareness interventions. As such, a PRISMA (preferred reporting items for systematic reviews and meta-analyses) systematic literature review was conducted to identify the theories that could inform the development of such an intervention. Based on the review, a theoretical framework was developed that incorporated the basic elements identified as guiding principles for the development of awareness interventions. In the same way, theory, specifically SAT was used to inform the development of an instrumental framework for designing context-sensitive awareness interventions, incorporating the basic elements that practitioners can use to develop context-sensitive awareness interventions. The findings from the study indicate that the social context, which encompasses the social issue at hand; available media content; the proximal and distal environments; and the choice of awareness tool, which is informed by the marketing mix (i.e., price, product, people, promotion, and place), affect the reach or distribution of the intervention. The theoretical framework contributes to the body of knowledge on the subject, while the instrumental framework provides a practical approach to building customizable interventions that can be tailored to specific contexts and available resources for awareness.

## 1. Introduction

Information technology artifacts and tools have benefitted many industries and homes due to their commercial and domestic utility, respectively. Information technology systems or information systems have been gradually integrated into society as key resources for efficient productivity and effective communication [1,2]. Information systems are also at the heart of the fourth industrial revolution (4IR). However, many other technologies have come before information technology, and many revolutions before the 4IR. For example, consider the first industrial revolution, which taught its participants how to labor using industrial machinery in steam-powered factories [3]. The second industrial revolution taught people how to read, write, and utilize mass production to maximize manufacturing [4]. The third taught people how to use, misuse, and abuse computers and internet connectivity in the name of digitization [4,5]. The fourth is teaching an over-reliance on information systems in the form of artificial intelligence, machine learning, automation, cloud computing, and big data industries [6,7].

With each industrial revolution, people either participate and gain a skillset or are left behind by exclusion. Technology in education has migrated learning activities online and encouraged learners to spend more time on their digital devices, thus increasing the rate at which children access online platforms [6]. With the high rate of information technology use, the current generation of youth are referred to as digital natives, as they were born in the technology era, thus for them, technology use is expected to be innate [8]. Without current technology skills, learners are perceived by their peers to be ignorant, uncool, at a social disadvantage, and thus become more likely to be involved in bullying activities online [9]. The move towards the fourth industrial revolution has driven a significant amount of technology user traffic to the internet, including children [6,10]. With daily social activities such as gaming and learning taking place online, children have also learnt to socialize online. In this context, the playground has thus moved online. Similarly, the challenges of the playground, such as bullying, have also moved online [8]. Hence, cyberbullying and mobile bullying have become an ongoing concern.

### 1.1. Mobile Bullying

Bullying is a form of aggressive behavior or harassment that involves repeated exposure to negative actions by one or multiple individuals over time, with the intention of causing harm or injury. This behavior occurs between individuals with unequal power dynamics [8,11]. Cyberbullying is a type of bullying that takes place online, and it can include acts such as intentionally excluding someone from online activities, the theft and distribution of personal information such as passwords, identity theft, and online harassment to intentionally cause harm to another person’s relationships or social standing [12,13]. Cyberbullying extends to repeated intentional harm inflicted by using the internet or electronic devices to insult or threaten others [14,15]. While cyberbullying is often considered an extension of conventional bullying, it differs in that it uses internet-connected electronic communication media [16]. Moreover, mobile bullying is a subset of cyberbullying that specifically focusses on the use of mobile devices to bully others [17,18]. The main difference between mobile bullying and other forms of bullying is the platform on which the behavior occurs. Mobile bullying can manifest in various ways, such as rude or offensive text and voice conversations, gossiping, chat rooms, social exclusion, and unauthorized photo distribution [19].

The adolescent population is the most involved and affected by mobile bullying with research indicating that in South Africa, youth between the ages of 12–22 years are the most victimized in terms of the mobile bullying type of peer aggression [20]. Mobile bullying incidents continue to rise, especially among adolescents [21,22]. The continued rise in mobile bullying incidents has led to a plethora of interventions developed and rolled out as a reactive response, with no consensus on what the key success factors or performance indicators should be. Several meta-analyses have been conducted regarding anti-bullying interventions with a focus on the ones that were considered to have been effective [23]. However, there is still no guiding framework for the development these types of interventions for raising awareness or any requirements to guide the development of an anti-mobile bullying intervention, more specifically in the African context. Research into the prevention of bullying has identified awareness as one of the most important areas that can reduce bullying in schools. In addition, Cilliers [24] further asserts in recent findings that in the context of South Africa, there is an overarching lack of awareness coupled with a lack of responsiveness. Thus, the current study aims to fill the research gap on the lack of awareness and responsiveness on mobile bullying. Thus, the current study worked towards developing a framework to guide the development of awareness interventions and piloting it in the context of mobile bullying awareness.

### 1.2. The Current Study

It may be necessary to explore intervention characteristics which may provide an understanding of how interventions affect/relate to awareness and bullying involvement [25,26]. Certain features of an intervention could be instrumental in raising awareness, reducing incidents or the severity of the incident outcomes. Therefore, this study focused on exploring the development of a useful anti-mobile-bullying intervention by gaining an understanding of what key elements are necessary for the process. It might be worth investigating if the same mobile technology platforms that are used for bullying could be useful in raising awareness and reducing bullying involvement among adolescents. Thus, we investigate information systems as potential tools for delivering awareness interventions.

It has been estimated that one in five adolescents reported to have been cyberbullied, with 25% citing numerous incidences [27]. Online platforms offer adolescents the ability to land painful blows on others from a place of safety, namely physical distance and anonymity [28]. This study aimed to develop a framework for building awareness interventions; thus, the researcher turned to the literature for theories that explain the phenomenon of awareness. Awareness is the development of mindfulness and acknowledging that an issue or topic exists [29]. The process of raising awareness is closely related to learning such that awareness requires learning [30]. Learning is the process of acquiring a skill as an outcome of a process. A person can only become aware of an issue after having learnt about that issue. The learning process does not have to be formal, structured, or intentional to develop awareness [8]. An individual’s failure to observe, comprehend, learn from, and or act on the elements of their environment to project possible future events and plan future actions constitutes a lack of awareness—specifically, a lack of situational awareness.

Situational awareness is defined as the ability of an individual to observe, comprehend, learn from, and act on the elements of their environment or situations to project possible future events in the same environment [31,32]. The author turned to the situational awareness theory to gain an understanding of how to raise awareness. The situational awareness theory (SAT), as coined in [33], has three overarching principles: the perception of the environment and its dynamics—such as the online environment; the comprehension of the current environment and how it affects an individual’s goals, including a worldview of the individual; and the projection and anticipation of potential future events [34,35]. SAT highlights that a person’s immediate social environment and the level of attention they give to what is going on around them could affect their awareness and behavior. This social context encompasses the social issue at hand, available media content, as well as the proximal and distal environments.

The SAT could be useful in explaining the essential elements of raising awareness by informing the questions to ask when building awareness models. As such, the SAT was used as the theoretical underpinning for this study. The necessary questions to ask in order to create situational awareness of an intervention’s contents, in the social context of the intervention, include the following:Who is the *target audience*—Who is affected?What is the *awareness context*—Where can the audience be reached?What is the *communication medium/tool* for awareness—What are the available means to reach the audience?What is the *awareness content or message*—What should be communicated in the awareness campaign?What are the *intervention timelines*—When should the target audience be reached? When is the best time to reach the target audience through the chosen medium?

See Figure 1 for the responses in the model related to developing an intervention for mobile bullying awareness in high schools. The arrows show the sequence of the considerations, starting with the target audience. 

Most research on cyberbullying and mobile bullying has focused on behavioral theories and learning theories to help explain behaviors and how new ones can be learnt, mostly in proximal and distal environments [37]. However, very little has been done to inform the process of developing awareness interventions.

After gaining an understanding of mobile bullying and awareness, the author sought concepts and theories from the fields of communications and marketing to inform the effective mass distribution of awareness messages. Raising awareness can be likened to a form of marketing strategy to distribute awareness messages [38,39]. The five Ps of marketing, also known as the marketing strategy mix, are product, place, promotion, price, and people [40,41]. These are the key elements of a marketing mix and are used to develop a marketing strategy. Each of these elements can be applied to awareness interventions when raising mobile bullying awareness, in the same way they are applied to brand awareness. The marketing mix could thus provide an understanding of how to better raise awareness.

*Product*: In the context of raising mobile bullying awareness, the product would be an awareness campaign or initiative. This includes identifying the specific message that needs to be communicated, the target audience, and the desired outcome. The product should be designed to resonate with the target audience and address the specific issue of mobile bullying [41]. The product refers to the thing which the marketing consumer is being informed about. In the context of raising awareness, the product would be represented by the awareness message about a topic or concept. The product (awareness topics presented to participants) could affect mobile bullying awareness in the participant population. And thus, the contents of the awareness message may affect the levels of awareness amongst adolescents. Therefore, the awareness message about mobile bullying can help raise awareness.*Place*: The place element of the marketing strategy mix refers to the distribution channels used to reach the target audience. For mobile bullying awareness interventions, this may include social media platforms, mobile apps, and other online channels where the target audience is most active [40]. The place, in the case of mobile bullying awareness, could represent the awareness context, i.e., the place where the target population can be reached. For this study, the context is the school environment. Thus, the social environment (home environment) may affect the awareness in the participant population.*Promotion*: Promotion includes the communication channels used to promote the mobile bullying awareness campaign. This includes social media advertising, email marketing, public relations, and other forms of promotion that are designed to generate awareness and engage the target audience [42]. The promotion aspect of an awareness intervention is the platform or medium where the awareness message is made available. The platform or medium of delivery can help raise awareness. Therefore, the medium or tool of delivery (i.e., awareness platform or channel as the intervention environment) may affect the awareness levels in the participant population [37].*Price*: The price element of the marketing strategy mix refers to the cost associated with the awareness campaign. This may include the cost of producing and distributing promotional materials, as well as any costs associated with running online ads or social media campaigns [41,42]. The price pertains to the cost of raising awareness. Practitioners may incur costs in providing awareness content. How much they are willing to spend on the intervention may affect the quality of the awareness intervention and subsequently affect the intervention’s effectiveness. Therefore, the price or costs may affect awareness levels among adolescents. Since the intervention is at no cost to the school and learners, the intervention development costs may affect the distribution and available features (tool complexity and reach) and the extent to which the tool becomes available to raise awareness.*People*: The people element of the marketing strategy mix refers to the people who are responsible for creating and executing the awareness campaign. This includes the project team, volunteers, and any external partners or stakeholders who are involved in the initiative [40]. The people represent the target audience for an awareness intervention. The participation rate of the people affects the number of people exposed to the intervention and thus affects the awareness levels of the population.

The marketing mix has previously been used to raise awareness of healthy lifestyle choices or healthy behavior. The use of the marketing mix to raise awareness is depicted in the conceptual framework showing the environmental advantage applied to market the type of transport used for goods [43]. Similarly, the marketing mix could help provide an understanding of how to increase awareness when the five Ps are included in an awareness intervention. Each of the five Ps can contribute to raising awareness as part of the tool (see Figure 2), i.e., the proposed conceptual framework used for the development of awareness interventions.

When effectively raising awareness using a digital intervention, both the literature and situational awareness theory show that environmental factors, the social issue at hand (inclusive of the social roles), and the media content of the awareness message have a direct impact on the effectiveness of raising awareness of an issue in a specific social context, as seen in Figure 2. Furthermore, the five Ps of marketing (price, product, people, promotion, and place) are instrumental in raising awareness [41,44]. Thus, raising awareness is dependent on the four aspects as constructs.

For the *social issue*: The social issue should be informed by the related social theory on how to effect awareness for that issue. In the context of mobile bullying, there are roles associated with those involved or affected [45,46]. The roles that emerge as a result of the social issue in question, i.e., mobile bullying, need to be considered when raising awareness, as per role theory and situational awareness theory [30]. Thus, role theory can shed some light on understanding the social roles that affect mobile bullying awareness in the participants’ social context.Regarding the *environment* as an aspect of the framework: The proximal and distal environments have an impact on raising awareness of a social issue [30,47]. Thus, for this study, immediate environments (school, home, online) and distal environments (government departments, legislations) have to be considered when developing interventions for communicating awareness. Distal environments regulate the educational content that is suitable for consumption by school-aged children [48].Regarding *media content*: The media richness theory informs the decision to consider the most appropriate media to communicate a message based on the needs of the audience [49]. Thus, the media richness theory provides a guide on what researchers can consider as an appropriate communication medium to use for communicating a message, given their audience. In recent years, instant messaging has become the most commonly used form of communication for daily use. As such, it is even embedded in corporate communication channels [50,51].For the *awareness tool*: On the left-hand side of the conceptual framework, the awareness tool and the messages it delivers are informed by the marketing mix which encompasses the five Ps of marketing (price, product, people, promotion, and place) to ensure that the awareness message is impactful to the intended audience [41,42].*Context-sensitive awareness intervention*: The elements of the framework that appear on the left-hand side are the independent variables, which influence the dependent variable that appears on the right-hand side of the framework in the diagram in Figure 2. The context-sensitive awareness intervention is informed by the social context, Where the issue of social concern is experience, as well as the proposed suitable tool for raising awareness in the target population during the process of intervening [41]. As such, an awareness intervention that is developed by following the proposed intervention conceptual framework should be suitable for the issue of interest and the social context. Thus, the intervention itself is dependent on the context and selected method of raising awareness, referred to in the model as the awareness tool [30].

## 2. Materials and Methods

A systematic review of the literature was conducted to develop a conceptual framework and to model an instrumental framework that is based on the identified theory, specifically the situational awareness theory [52]. The steps outlined below, as described in [8], were followed in order to obtain a systematic literature review for this study (see Figure 3):

6.Definition of the search protocol: The sources for the literature were online academic databases. The search terms for the study were (“*awareness interventions”); (“*bullying awareness*”); (“cyberbullying intervention*”); (“awareness” or “aware*”); and (“*bullying*” AND “*intervention*”).7.Search and identification of academic papers: A search was conducted on academic databases (Google Scholar, Web of Science, and Scopus) for publications using the keywords defined in the search protocol. The period of the review was 15 years and included publications between the years 2009 and 2024. A total of 327 academic articles resulted from the search. During this period, there was a rapid increase in the social use of digital devices. Moreover, there was a rise in cyberbullying and mobile bullying.8.Paper selection and screening: From the search results of 327 publication records, 19 duplicate entries were removed and 308 research papers were screened. From the screening process, a total of 272 records were excluded based on the exclusion criteria, i.e., 245 entries had a focus on bullying involvement and 27 on cyberbullying legislation instead of intervention development.9.Review and analysis of the selected papers: After the screening process, 36 entries remained, of which 21 were excluded as they did not involve any awareness interventions. Thereafter, 15 publications remained, of which 7 were excluded on the basis that they focused on law enforcement. A total of 8 papers focused on the development of awareness interventions and were analyzed for this study (see Table 1).10.Result presentation and write-up: A review of the applied theories was conducted to develop an instrumental model as well as a conceptual framework in order to guide the development of an awareness intervention.

**Table 1 ijerph-22-00774-t001:** The eight (8) sampled and reviewed articles for the study.

Study Location	Objective	Methodology	Intervention	Key Outcomes
1. Western Australia [12]	To review existing whole-school interventions and inform cyberbullying intervention strategies.	Meta-analysis	None. Review of existing interventions.	In school-based anti-bullying programs, whole-school interventions are the most effective. Bullying can be prevented and managed through non-stigmatization. Six broad whole-school indicators were included.
2. Campania, Italy [13]	To evaluate the Tabby Improved Prevention and Intervention Program (TIPIP) and its long-term effectiveness.	Experiment	Tabby Improved Prevention and Intervention Program for cyberbullying and cybervictimization.	The study found that the TIPIP intervention was ineffective or had limited effectiveness in the long term, despite significant effectiveness in the short-term post implementation.
3. Perlis State, Malaysia [15]	To review the use of a multimedia application (CyBA) to increase cyber-bullying knowledge and to evaluate the effectiveness of the CyBA intervention.	Quasi-experimental research design	Multimedia application for cyber-bullying awareness (CyBA).	The study results indicate that perceived knowledge and awareness increased after participant interaction with the intervention. Thus, they concluded that the CyBA intervention had a positive impact among adolescents.
4. South Africa [17]	To descriptively present the development and implementation process for a mobile application as an intervention to monitor and report mobile victimization in high schools.	Design science research	Mobile application for monitoring and reporting victimization, based on the Mobile Victimization Typology (MVT).	The design science process of developing the digital application was documented and explained. The development followed the Mobile Victimization Typology (MVT) structure to improve the process of identifying, describing, explaining, and comparing mobile victimization.
5. United Kingdom [22]	To determine the effectiveness of existing school-based anti-bullying programs in reducing school-bullying behaviors.	Meta-analysis	None. Review of existing interventions.	The study found that school-based antibullying interventions were effective in reducing bullying perpetration by 18–19%, and victimization by about 15–16%.
6. Regional: Europe, North America, and Scandinavia [23]	To review the effectiveness of of school-based bullying prevention programs.	Meta-analysis	None. Review of existing interventions.	The NoTrap! Program had significant effectiveness in reducing bullying victimization. The Olweus Bullying Prevention Program had the largest effects regarding bullying perpetration outcomes.
7. Europe [24]	To investigate the design and implementation insights of two European projects aimed at supporting schools with the training of resilience as a protective factor to prevent (cyber)bullying and with the deployment of innovative digital interventions to detect and prevent this phenomenon early.	Co-design	UPRIGHT program and CREEP Virtual Coach. CREEP Semantic Technology is a detection tool for cyber threats in schools.	The main challenges and insights collected during the design and implementation of both interventions are discussed to inform future research and practice. Conclusion: The feasibility and acceptance of prevention programs are key to reducing the risk of (cyber)bullying and improving the psychological well-being of young adolescents.
8. [38]	To systematically review the literature, the authors identified public-directed interventions to promote awareness of antibiotics prescriptions.	Systematic review	Antibiotic awareness.	Many interventions were multifaceted and distributed educational materials within the community as well as clinical sites. Most studies that measured antibiotics prescriptions reported success.

The PRISMA (preferred reporting items for systematic reviews and meta-analyses) flow diagram in Figure 3 depicts the research process, showing steps 2, 3, and 4 [53]. The next section presents the results from the systematic review of the literature that was conducted for the current study.

## 3. Results

The output from the systematic literature review is presented in Table 1. The headings indicate the review focus for the final eight articles that were analyzed. The results indicate the study location, with the citation, article objective, research methodology, and intervention name and type, as well as the overall key outcomes from the research article.

From the results of the PRISMA literature review, it is evident that a significant number of the key outcomes included intervention effectiveness. However, effectiveness can only be determined when an evaluation has been conducted. Hence, the evaluation and effectiveness aspects have been included to extend the conceptual framework of Figure 2 into Figure 4.

The refined framework includes effectiveness as a key element of the final intervention. After the successful development of the context-sensitive awareness intervention, the intervention should be tested for its effectiveness. Thus, it is important for the intervention to be testable, so that the effectiveness evaluation can be conducted during the pilot and or implementation rollout stages.

## 4. Discussion

A majority of studies on bullying and interventions used behavioral theories and learning theories and focused on the types and levels of aggression in cyberbullying. This study consulted and used marketing and communications theories to focus on the instrumental development of the awareness tools which would be used to achieve mobile and cyberbullying interventions.

The primary objective of this research was to develop a systematic process that informs the development of awareness interventions. The process was encapsulated in a comprehensive framework that could facilitate the development of mobile bullying awareness interventions in the appropriate context. To accomplish this, a conceptual framework was devised that incorporated fundamental principles that are crucial for the development of awareness interventions. Similarly, theory, specifically SAT, was used to inform the development of an instrumental framework that can be utilized by practitioners to design context-sensitive awareness interventions. The findings of this study suggest that the social context, which takes into account the social issue at hand, the available media content, the proximal and distal environments, and the choice of awareness tool (which is informed by the marketing), plays a significant role in determining the reach or distribution of the intervention. The conceptual framework contributes to the existing body of knowledge on the subject, while the instrumental framework presents a practical approach to creating testable, customizable interventions that can be adapted to specific contexts and resources for raising awareness.

## 5. Conclusions 

Cyberbullying has become common among adolescents, especially when they have access to social media platforms. Studies suggest restricting child access to mobile devices until a certain age. However, this exclusion would also exclude those children from online learning activities. With the rise in cyber and mobile bullying, there is an increase in studies aiming to understand, explain, and contain the scourge of online bullying incidents. This was also the rationale for the present study, which aimed to create a guiding framework for developing awareness interventions. This study revealed that the social context, which encompasses the social issue at hand, the available media content, the proximal and distal environments, and the choice of awareness tool, as informed by the marketing mix (i.e., price, product, people, promotion, and place), affects the reach and distribution of the intervention. Thus, it is essential to ask probing questions which allow an intervention to use the resources available in a research context to develop an awareness intervention that is accessible and appropriate to the social issue and its context. 

## Figures and Tables

**Figure 1 ijerph-22-00774-f001:**
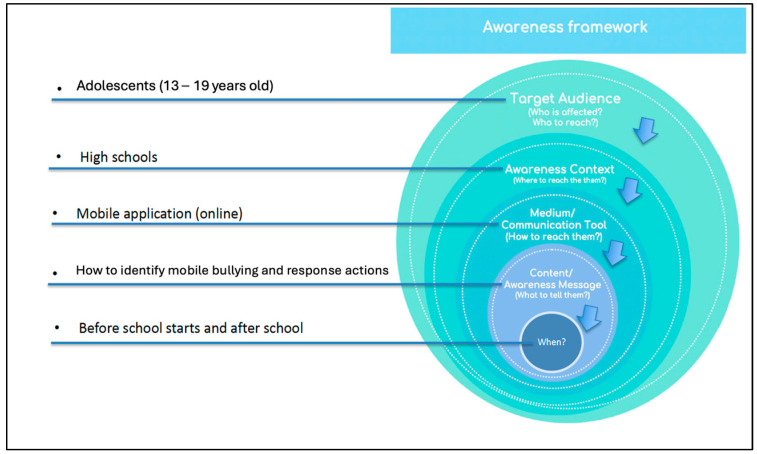
Instrumental framework for the basic requirements of awareness interventions [36].

**Figure 2 ijerph-22-00774-f002:**
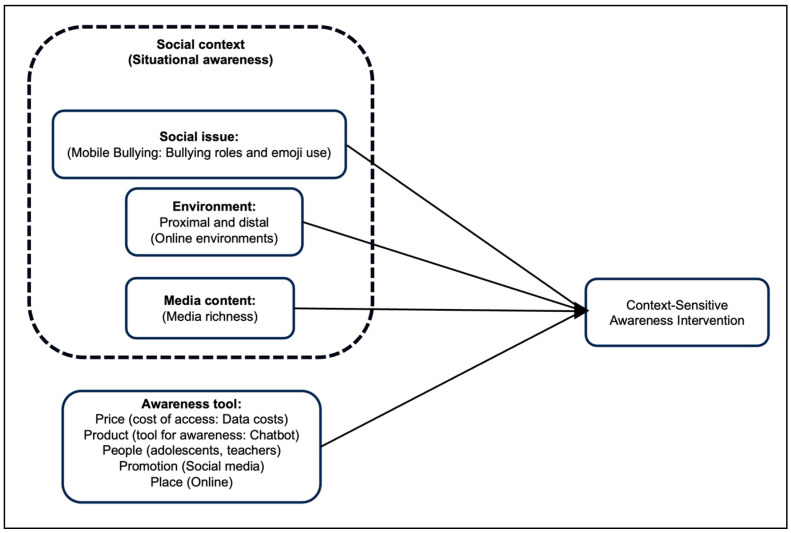
A conceptual framework for developing awareness interventions.

**Figure 3 ijerph-22-00774-f003:**
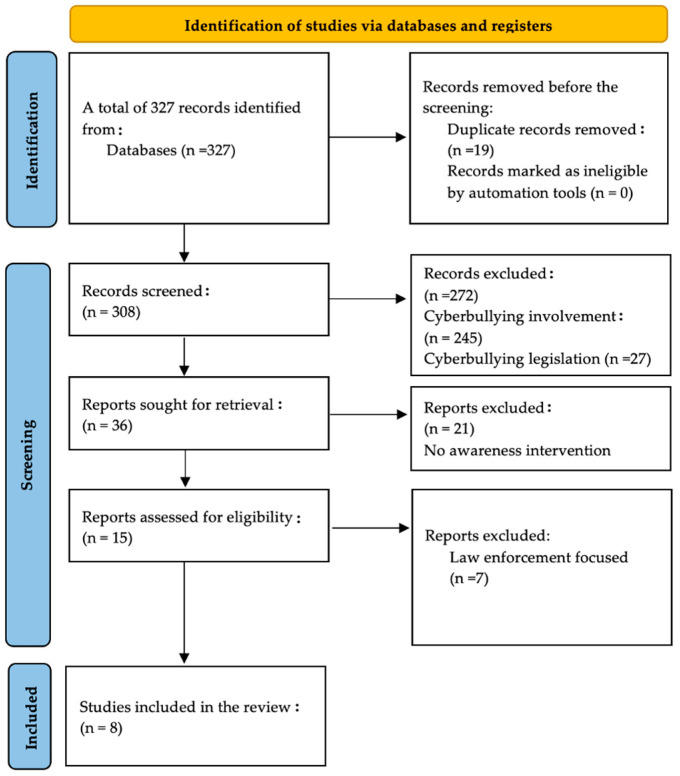
PRISMA flow of the research method, adapted from [36].

**Figure 4 ijerph-22-00774-f004:**
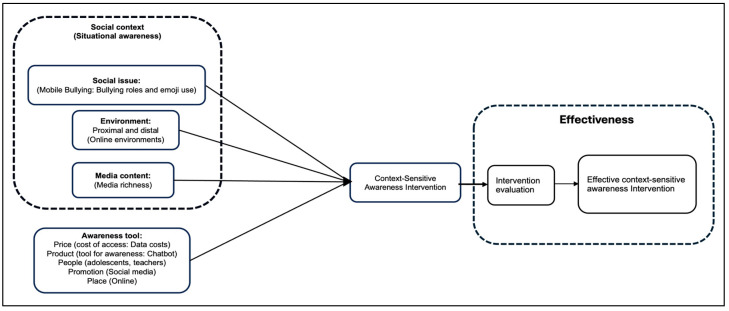
A refined conceptual framework for developing awareness interventions.

## Data Availability

No new data were created or analyzed in this study.

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
