# Peer review of "A Framework for Developing Awareness Interventions: A Case of Mobile Bullying"

_ijerph, 2025, doi:10.3390/ijerph22050774_

Round 1

Reviewer 1 Report

Comments and Suggestions for Authors

I would like to thank the Editor for the opportunity to review this study and I am flattered to be able to make my contribution. This work has left me with a bittersweet feeling. It addresses a very important and relevant topic, but it is, at the same time, very confusing as it is organized and written. From my point of view, I think some changes are necessary:

(a) The title does not exactly respond to the content. I think it should refer to the systematic review as a basis for developing awareness in mobile bullying intervention.

b) The abstract should be rewritten, referring to the systematic review and not to an exploratory and literature review. It should be organized around objectives, methodology, results and proposal, and discussion and conclusions.

c) Methodology: The authors should specify the period in which the literature review was carried out (years) and indicate in the bibliography with an * those included in the review.

d) The objective(s) should be clarified, as it is confusing.

e) In the results section, the 8 studies included in the review should be analyzed in depth. It is usually useful to include a table where the articles are analyzed in depth (objectives, sample, instruments, sessions, etc.), in order to propose a framework for developing awreness interventions.

f) In this same results section, a subsection could be included where, based on the results of the systematic review, a framework for developing awreness interventions could be developed.

Author Response

The comments were attended to and addressed. Thank you for the in-depth and insightful feedback. It has enriched the study.

Review comments

Action

a)       The title does not exactly respond to the content. I think it should refer to the systematic review as a basis for developing awareness in mobile bullying intervention.

a) Comments addressed. The title was revised from: A Framework for Developing Awareness Interventions: A Case of Mobile Bullying.

To: Building a Mobile Bullying Intervention: Insights from a Systematic Review.

b)      The abstract should be rewritten, referring to the systematic review and not to an exploratory and literature review. It should be organized around objectives, methodology, results and proposal, and discussion and conclusions.

b) The abstract was updated to refer to the systematic review. The comments about structure were noted. The structure is per the journal requirements and logical flow of the project execution.

c)       Methodology: The authors should specify the period in which the literature review was carried out (years) and indicate in the bibliography with an * those included in the review.

c) The literature review study period was added in section 2, item 6.

The literature review articles that were included have been indicated in the Bibliography with a *.

d)      The objective(s) should be clarified, as it is confusing.

d) The objectives were clarified and elaborated in more detail.

e)       In the results section, the 8 studies included in the review should be analyzed in depth. It is usually useful to include a table where the articles are analyzed in depth (objectives, sample, instruments, sessions, etc.), in order to propose a framework for developing awareness interventions.

e) In the results section, the 8 studies included in the review were analyzed in depth and presented in table format. Thank you. This was helpful in providing depth to the study.

f)        In this same results section, a subsection could be included where, based on the results of the systematic review, a framework for developing awareness interventions could be developed.

f) The framework is now presented after the analysis in the results section.

Reviewer 2 Report

Comments and Suggestions for Authors

Comments for the author:

I enjoyed reading this paper, and it was well-written and explained. However, there are some minor edits that I suggested for it.

Abstract:

The aim of the study, as suggested by the author, is to develop a framework for increasing awareness. However, the research gap was not mentioned. Lines 82-86 are about the existing gap. I suggest adding a brief gap in the abstract as well.

Introduction

There is a good narrative. However, I suggest adding a section for bullying to mention different types of bullying, its causes, and the most recent findings of its effects. Then, there is a section for mobile bullying or cyberbullying.

Throughout the introduction, I did not understand why the authors developed an awareness intervention. And no information about how awareness could be helpful in this matter! Lines 100-104 belong to the introduction, not the problem/research statement. Also, theories behind awareness and types of awareness should be added. Also, what kind of awareness was focused on in this intervention?

Line 108 introduces Situational awareness theory! Where was this before the methods? Add it to the introduction. Also, there is no theoretical framework for your study! A theory must guide your study. You could use situational awareness theory as your theoretical framework, but you need to explain how this theory supports your aim. This section should be the first paragraph in the “Current Study.”

In methods, please indicate which authors screened the data. Instead of explaining why some studies were excluded, add a paragraph about the inclusion and exclusion criteria.

Also, please provide the quality assessment of the studies you have included!

It is advisable to prioritize the scientific information in lines 83-85, rather than delving into the authors' intentions and goals. This approach can enhance the study's clarity and comprehensibility for readers. Maintaining coherence and emphasizing clear results is crucial. Reducing such statements can promote better understanding and comprehension of the results.

Author Response

The comments were attended to and addressed. Thank you for the in-depth and insightful feedback. It has enriched the study.

Review comments

ActionActions

Abstract:

The aim of the study, as suggested by the author, is to develop a framework for increasing awareness. However, the research gap was not mentioned. Lines 82-86 are about the existing gap. I suggest adding a brief gap in the abstract as well.

Abstract:

The research gap (lines 82-86) was added to the abstract. About the lack of frameworks to guide the development of awareness interventions.

“There is a gap from the lack of frameworks and requirements to guide the development of awareness interventions”.

Introduction

There is a good narrative. However, I suggest adding a section for bullying to mention different types of bullying, its causes, and the most recent findings of its effects. Then, there is a section for mobile bullying or cyberbullying.

Throughout the introduction, I did not understand why the authors developed an awareness intervention. And no information about how awareness could be helpful in this matter!

Introduction

Mobile bullying definitions included in section 1.1.

The research gap (now added to the abstract per your guiding insight) is on guiding principles for awareness interventions. The study provides the fundamental considerations to guide practitioners on the process of developing awareness interventions. The focus is on the instrumental process, not the type of intervention (awareness intervention).

Lines 100-104 belong to the introduction, not the problem/research statement.

Lines 100-104  moved from the problem/research statement section to conclude the introduction section. Thank you.

Also, theories behind awareness and types of awareness should be added. Also, what kind of awareness was focused on in this intervention?

Line 108 introduces Situational awareness theory!

The Situational Awareness Theory (SAT) and types of awareness added; and expanded on Line108.

Where was this before the methods? Add it to the introduction. Also, there is no theoretical framework for your study! A theory must guide your study. You could use situational awareness theory as your theoretical framework, but you need to explain how this theory supports your aim. This section should be the first paragraph in the “Current Study.”

The SAT discussed before the methodology as the theoretical underpinning for the study in section 1.2- The current study. The type of intervention is added at the end of section 1.1.

In methods, please indicate which authors screened the data. Instead of explaining why some studies were excluded, add a paragraph about the inclusion and exclusion criteria.

Individual author roles highlighted for the data screening component.

Also, please provide the quality assessment of the studies you have included!

The methods section updated to reflect the authors that conducted the data screening. The inclusion and exclusion criteria were also highlighted in the methods section.

Quality assessment of the study is in the PRISMA inclusion criteria and the results table of the literature review.

It is advisable to prioritize the scientific information in lines 83-85, rather than delving into the authors' intentions and goals. This approach can enhance the study's clarity and comprehensibility for readers. Maintaining coherence and emphasizing clear results is crucial. Reducing such statements can promote better understanding and comprehension of the results.

Emphasis on clear results maintained as much as possible.

Round 2

Reviewer 1 Report

Comments and Suggestions for Authors

The authors have adequately addressed the issues raised.